PREREGISTERED RESEARCH ARTICLE

# Removing facial features from structural MRI images biases visual quality assessment

Céline Provins[1], Élodie Savary[1], Thomas Sanchez[1,2], Emeline Mullier[1,3], Jaime Barranco[1,2,4,5], Elda Fischi-Gómez[1,2,6], Yasser Alemán-Gómez[1,7], Jonas Richiardi[1,2], Russell A. Poldrack[8], Patric Hagmann[1], Oscar Esteban[1]*

1 Department of Radiology, Lausanne University Hospital and University of Lausanne, Lausanne, Switzerland, 2 CIBM - Center for Biomedical Imaging, Lausanne, Switzerland, 3 EEG and Epilepsy Unit, University Hospital and Faculty of Medicine of Geneva, University of Geneva, Geneva, Switzerland, 4 School of Engineering, Institute of Systems Engineering, HES-SO Valais-Wallis, Sion, Switzerland, 5 The Sense Innovation and Research Center, Lausanne and Sion, Switzerland, 6 Signal Processing Laboratory (LTS5), École Polytechnique Fédérale de Lausanne (EPFL), Lausanne, Switzerland, 7 Center for Psychiatric Neuroscience, Department of Psychiatry, Lausanne University Hospital and University of Lausanne, Prilly, Switzerland, 8 Department of Psychology, Stanford University, Stanford, California, United States of America

* phd@oscaresteban.es

## Abstract

A critical step before data-sharing of human neuroimaging is removing facial features to protect individuals' privacy. However, not only does this process redact identifiable information about individuals, but it also removes non-identifiable information. This introduces undesired variability into downstream analysis and interpretation. This registered report investigated the degree to which the so-called *defacing* altered the quality assessment of $T_1$-weighted images of the human brain from the openly available "IXI dataset". The effect of defacing on manual quality assessment was investigated on a single-site subset of the dataset ($N$ = 185). By comparing two linear mixed-effects models, we determined that four trained human raters' perception of quality was significantly influenced by defacing by modeling their ratings on the same set of images in two conditions: "nondefaced" (i.e., preserving facial features) and "defaced". In addition, we investigated these biases on automated quality assessments by applying repeated-measures, multivariate ANOVA (rm-MANOVA) on the image quality metrics extracted with *MRIQC* on the full IXI dataset ($N$ = 581; three acquisition sites). This study found that defacing altered the quality assessments by humans and showed that *MRIQC*'s quality metrics were mostly insensitive to defacing.

## Introduction

The removal of facial features—or *defacing*—is necessary before sharing anatomical images of the brain to protect participants' privacy [1] in compliance with some local

**Data availability statement:** All the new materials relating to this work were shared under suitable open licenses (Apache 2.0 for code and CC-BY for data, unless otherwise specified) before the Stage 2 report submission. The IXI dataset is available at https://brain-development.org/ixi-dataset/ under the Creative Commons CC BY-SA 3.0 license. All the tabular source data—i.e., manual ratings and extracted IQMs,—as well as the outputs of analyses are made available with this manuscript as supplementary data files and under the same license as the manuscript itself. Specifically, files S1 Data and S2 Data contain the source data, files S3 Data, S4 Data, and S5 Data the PCA projections for each of the three corresponding analyses and finally, files S6 Data and onwards contain the analyses' outputs. The MRIQC reports generated from the IXI dataset and used by the human raters in their assessment have been deposited on FigShare (doi:10.6084/m9.figshare.28684367) under the terms of the CC-BY-SA-4.0 license, as they are derived from the original IXI dataset. The dataset is accessible under version control with DataLad at https://github.com/TheAxonLab/defacing-and-qc-ixi-reports. All the code related to this study is openly shared under the Apache 2.0 license at https://github.com/TheAxonLab/defacing-and-qc-analysis. The specific commit in the Git history of the repository containing the code to produce all the results included in this manuscript was tagged and deposited in Zenodo (doi:10.5281/zenodo.15103450). The presentation used in the training session can be accessed at https://osf.io/d57mb/, and the training reports are openly shared at https://github.com/TheAxonLab/defacing-and-qc-trainingsession. Should any of the participants' data be recalled from the original IXI dataset, e.g., after a UK GDPR request, we will accordingly recall the corresponding data derived from the participant's T1w image.

**Funding:** This work has been supported by the Swiss National Science Foundation—SNSF—(#185872, OE), the NIMH (RF1MH121867; OE, RP), and the Chan-Zuckerberg Initiative (EOSS5-266; OE). PH and YAG receive support from SNSF (#185897, PH), EFG receives support from SNSF (#10000706, EFG). The funders had no role in study design, data

privacy protection regulations, such as the General Data Privacy Regulation (GDPR) in Europe [2] or the Health Insurance Portability and Accountability Act (HIPAA) in the US [3]. Defacing is typically implemented by zeroing, shuffling, or filtering the content of image voxels located in an area around the participant's face and, often, the ears (see Fig 1). Defacing is a destructive step with the potential to alter the results of downstream processing. De Sitter and colleagues [4] investigated the undesired effects of defacing on automated workflows in neurodegeneration studies. Failure rates—i.e., the percentage of inputs for which workflows did not finish successfully—were substantially larger on defaced inputs (19%) compared to their nondefaced counterparts (2%). They also reported systematic differences between the same processing with and without defacing in several outcomes of interest, such as gray- and white-matter volume and cortical thickness. Schwarz and colleagues [1] showed how these failures propagate and accumulate downstream, leading to substantial changes in the study outcomes. In a similar approach to our design, Bhalerao and colleagues [5] explored the impact of different defacing tools on a subset of image quality metrics (IQMs) automatically generated with *MRIQC* [6]. They found that all defacing tools impacted a subset of IQMs, and they estimated corresponding effect sizes on a sample limited to 30 subjects with a univariate modeling approach. Moreover, they identified further effects on the downstream segmentation of images. To the best of our knowledge, no study has investigated the impact of defacing on the manual assessment of MRI data quality.

Understanding how defacing alters the initial quality checkpoint of the research workflow [7] is necessary because there is compelling evidence that data showing specific artifacts or insufficient overall quality introduce bias into the results of analyses, raising questions about their validity [8–10]. For example, Alexander-Bloch and colleagues [10] showed that in-scanner motion can lead to systematic and region-specific biases in anatomical estimation of features of interest, such as cortical thickness. Therefore, it is critical to reliably identify substandard MRI data and exclude them early from the research workflow (quality control, QC). Our previous efforts to automate QC by learning *MRIQC*'s IQMs [6,11] demonstrated that site-effects imposed the largest limitation on the performance. The reportedly modest reliability of automated QC alternatives currently available explains that QA/QC continues to require the screening of imaging data on a one-by-one basis. Because visual inspection is time-consuming and prone to large intra- and inter-rater variabilities, implementing assisting tools and protocols to screen large datasets efficiently is an active line of work (e.g., *MRIQC* [6], *MindControl* [12], and *Swipes4science* [13]). Large consortia have also made substantial investments in this critical task and have generated valuable contributions to quality assurance (QA) protocols and corresponding QC criteria, e.g., the Human Connectome Project [14] or the INDI initiative (QAP [15]). One related but conceptually innovative approach was proposed by the UK Biobank [16], where sufficient quality was operationalized as the success of downstream processing. Given the massive size of the UK Biobank, Alfaro-Almagro and colleagues [16] flagged the images that did not successfully undergo pre-processing for exclusion. Although image exclusions were related most often to

collection and analysis, the decision to publish, or the preparation of the manuscript.

**Competing interests:** The authors have declared that no competing interests exist.

**Abbreviations :** BA plots, Bland–Altman plots; CI, confidence interval; cnr, contrast-to-noise ratio; df, degrees of freedom; efc, entropy-focus criterion; FDR, false discovery rate; IPA, in-principle accepted; IQMs, image quality metrics; LME, linear mixed-effects; LoA, limits of agreement; PCA, principal components analysis; QA, quality assurance; QC, quality control; rm-MANOVA, repeated-measures multivariate ANOVA; SNR, signal-to-noise ratio; T1w, $T_1$-weighted; MRI, magnetic resonance imaging; ANOVA, Analysis of Variance; HH, Hammersmith Hospital; GH, Guy's Hospital; IoPPN, Institute of Psychiatry, Psychology and Neuroscience

qualitative issues on images (e.g., artifacts), some images were discarded without straightforward mapping to quality issues. Moreover, because manual QC is onerous, many teams have attempted automation, either by defining *no-reference* (that is, *no ground truth is available*) IQMs that can be used to learn a machine predictor [6,15,17] or by training deep models directly on 3D images [18]. However, the problem remains extremely challenging when predicting the quality of images acquired at new MRI devices yet unseen by the model [6,11].

In a recent exploration [19], we found preliminary evidence that defacing alters both the manual and automatic assessments of $T_1$-weighted (T1w) MRI on a small sample (*N* = 10 subjects), implemented with *MRIQC*. The present registered report built on our exploratory analysis and confirmed that four trained human raters made systematically varying QC decisions when rating the same images depending on whether they have been defaced. Regarding automated QC, this paper showed that differences in IQMs extracted with *MRIQC* before and after defacing were not meaningful.

## Methods

The lettered footnotes in this section indicate deviations from the Stage 1 IPA (in-principle accepted) manuscript—https://doi.org/10.17605/OSF.IO/QCKET—and are reported in the "Protocol deviations" subsection.

### Hypotheses

This pre-registered report was set out to confirm whether defacing alters the manual and automatic QC of T1w images of the healthy human brain—implemented with *MRIQC*. This overarching question was tested in two specific hypotheses:

1. Defacing influences trained experts' perception of quality, leading to significant differences in quality ratings between the defaced and the nondefaced images. Specifically, we expected raters to assign higher ratings—on average—in the defaced condition than in the corresponding nondefaced condition (see Fig 1).

2. Defacing influences automatic QA/QC with *MRIQC*, introducing significant and systematic biases in vectors of IQMs extracted from defaced and nondefaced images. As evidenced by our preliminary data [19], these biases may showcase one direction for some IQMs and the opposite or no effects on others. Therefore, a directionality of effects could not be hypothesized.

### Data

This analysis was based on the publicly available IXI dataset [20], which contains 581[a] nondefaced T1w images acquired at three different sites in London (UK). One site was the Hammersmith Hospital (HH in the following), which used one 3 Tesla (3T) scanner. The two other sites (Guy's Hospital and Institute of Psychiatry, Psychology and Neuroscience; GH and IoPPN, respectively) featured 1.5T devices. The scanner parameters available for each site are listed in Table A in S1 Text. None of

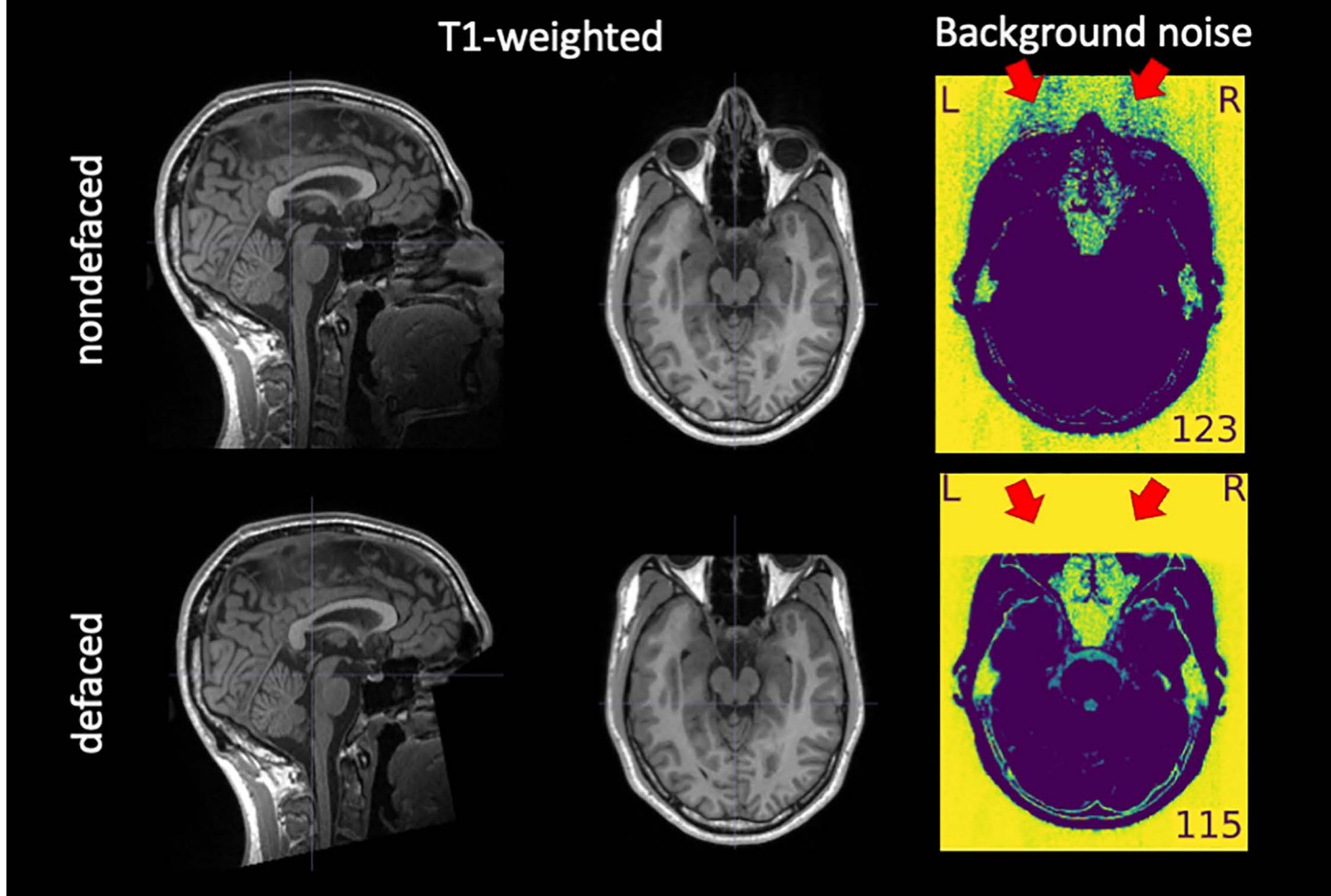

**Fig 1. An example of T1w image before and after defacing.** Defacing is typically implemented by zeroing the voxels around the face (left-hand side, panel "T1-weighted"). The "Background noise" panel shows two visualizations extracted from the *MRIQC* visual report, in which a window is applied to select the lowest intensities and then the latter are inverted to enhance patterns in the background. The red arrows indicate aliases along the anterior–posterior axis—which had the lowest bandwidth in the example—produced by eye motion. These aliases are straightforward to notice in front of the ocular globes in the "Background noise" panel of the "nondefaced" image because of the absence of other signal sources. This aliasing also spreads in the opposite direction, overlapping brain tissue. However, this overlap is often very hard to notice against the signals of interest within the brain. The corresponding "defaced" version of the "Background noise" panel shows how defacing eliminates valuable information for quality assessment.

the authors had screened or queried the dataset before pre-registration to anticipate quality-related patterns or summary statistics. Moreover, except for author OE, authors neither accessed nor performed any processing on the data before pre-registration. To investigate Hypothesis 1, the initial inclusion criterion was "all subjects with a T1w scan acquired at the 3T site (HH)", yielding a total of 185 subjects[b]. In testing Hypothesis 2, we included all participants in the IXI dataset with a T1w scan (581[a] subjects)[b]. Images were excluded from the evaluation of hypotheses 1 and 2 in the case of complete failure of image reconstruction[b]. The local ethics committee (Swiss BASEC—Business Administration System for Ethics Committees) has approved the processing of nondefaced images (project 2022-00360). This study involved only the reuse of publicly shared data. The original data were collected in accordance with the Declaration of Helsinki, with written informed consent obtained by the original investigators. This study did not attempt to re-identify the participants nor facilitate in any way such efforts[c].

**Data processing.** First, a defaced version of each scan ($N$ = 581) was generated with *PyDeface* [21]. We chose *PyDeface* because it currently presents the highest success rate at removing facial features while not removing brain voxels [22]. Furthermore, Bhalerao and colleagues [5] showed that *PyDeface* resulted in the smallest effect size on a subset of *MRIQC*'s IQMs. Fatal failure of *PyDeface* was the only exclusion criterion. Because all images were successfully processed, no images were dropped from the analysis for this reason. To avoid exclusion bias that could artificially inflate the effect size, images showcasing low-quality defacing, such as those partially retaining facial features—e.g., the eyes,—were included in the analysis[d]. These cases were expected to be more similar between defaced and nondefaced conditions, potentially attenuating differences in human ratings and IQMs. The raters then assessed the quality of the same images in two conditions (nondefaced and defaced) following the Manual assessment protocol described below. Raters did not have access to the mapping between defaced and nondefaced counterparts. We obfuscated participant identifiers and shuffled their ordering before presentation. The latest version in the 23.1 series of *MRIQC* (version 23.1.0) was then executed on all the T1w images available (that is, nondefaced and defaced). Then, the IQMs generated by *MRIQC*, corresponding to every image in the two nondefaced/defaced conditions, were collated and converted into a tabular format. Two subjects from the GH site (1.5T) were excluded because *MRIQC* failed to run on both the nondefaced and defaced images, reducing the dataset size to test Hypothesis 2 to 579 subjects. Hypothesis 1 (HH site) was not affected by this exclusion[e].

**Manual assessment protocol.** We performed manual quality assessment only on the images from the site with a 3T device (HH; $N$ = 185). This choice eliminated the field strength and other variability sources emerging from the specific scanning site as potential random effects. Moreover, images acquired with the 3T scanner were expected to showcase a signal-to-noise ratio (SNR) approximately twice as high as the SNR of images acquired with 1.5T scanners. Thus, the images acquired with the 3T scanner likely yielded, on average, better quality assessments by human raters independently of the defacing condition. Four human raters (authors TS, EM, JB, and EFG[f]) assessed the quality of the subsample in each of the two conditions (that is, defaced and nondefaced). The quality assessment was carried out by individually screening one *MRIQC*-generated visual report per subject and condition. These reports are openly shared (FigShare 10.6084/m9.figshare.28684367). Raters were recruited by inviting researchers who expressed their interest in QA/QC within the Department of Radiology of the Lausanne University Hospital (Lausanne, Switzerland) via e-mail. We did not impose restrictions on the raters' experience beyond familiarity with T1w images of the human brain. To ensure the consistency of their training, raters read our published QC protocol [23] and participated in a 3h[g] training session. Following our guidelines [23], raters were instructed that it was critical to scrutinize the background visualization carefully because most of the artifacts are more noticeable in the absence of signal of interest. Several examples supported this remark during the training sessions (see Fig 1, Fig U in S1 Text and the training materials). At the end[g] of this session, raters self-assessed their experience as either beginner, intermediate, or advanced.[h] One rater identified themself as an advanced rater, the three others as intermediate. The training session included an introduction to *MRIQC*'s visual reports, its "rating widget" (Fig A in S1 Text), and the data collection tools and settings. We also presented the list of quality criteria to consider, and for each criterion, we illustrated an exclusion example. These examples were extracted from diverse, publicly available datasets [24–27] and in-house datasets. Lastly, in the same session, the raters were asked to assess 20 images independently, and we compared and discussed the ratings together to ensure that the quality criteria were interpreted consistently. None of the training and calibration examples were extracted from the IXI dataset. The materials corresponding to the training session, as well as the self-assessments of experience, are openly shared for future exploration (see Data and code availability statement). To assess the intra-rater effects on QC, 40 subjects selected randomly were presented a second time in both conditions to all raters without their knowledge. This summed up to 450 images per rater (225 images per condition). We chose to repeat 40 subjects because it represented a good trade-off between having enough statistical power and the risk of having raters who would not complete their assignment. The random number generator to choose the 40 repeated subjects and the blinding of participant identifiers

were initialized with the timestamp of the registered report submission toward Stage 1. It was converted to an integer with the format YYMMDD + SSmmHH (Y: year, two last digits; M: month, D: day; S: seconds; m: minutes; H: hour). This seed was then preserved, clearly reported, and set for all the analyses. After screening each visual report, the raters assigned each image a quality grade with the rating widget presented in Fig A in S1 Text[h]. A quality score was assigned using a slider that permits the selection of numbers in a continuous scale from 1.0 to 4.0 (interval step of 0.05 and the value of 1.0 corresponding to the lowest quality). As presented in Fig A in S1 Text[h], the slider was presented with four categorical ranges for reference. The starting position of the slider was set in the middle. The raters were instructed to assess each subject according to our QC protocol [23], and they did not have access to the IQMs. All raters viewed the visual reports (FigShare 10.6084/m9.figshare.28684367) on a single LED panel of 43" screen diagonal, 3,840 × 2,160 resolution, a typical static contrast of 5,000:1, and the same ambient lighting. *MRIQC* reports feature a stopwatch that recorded the exact time each assessment took. The time for each assessment was measured and is available for future exploration. The removal of the IQMs from the visual reports, the assignment of images in the two conditions to raters, the shuffling of ordering of presentation, the actual presentation, and the tracking of raters' progress was managed with *Q'kay* [28], a Web Service we developed for this study.[b] Images assigned the lowest grade (one in our 1–4 interval scale) in both conditions by all raters would be excluded from all analyses. No images met this criterion.

## Experiments

**Determining that defacing biases the human raters' assessments of quality.** We tested the influence of the defacing condition and the rater (within-subject factor variables) on the ratings (dependent variable) by comparing linear mixed-effects (LME) model fits by means of a likelihood-ratio test. The LME modeling was a contingency alternative in case data violated some of the normality and sphericity assumptions of repeated-measures ANOVA (rm-ANOVA). As opposed to multiple *t*-tests, rm-ANOVA and LMEs enabled disentangling the inter-rater variability from intra-rater variability due to defacing and quantifying the latter. Because we do not necessarily expect the ratings distribution of each rater to have the same mean, rm-ANOVA and LMEs accounted for the baseline difference in ratings by adding the rater as a random effect in the model. We removed the 40 repeated images from the dataset because neither rm-ANOVA nor LMEs allow non-uniquely identified rows[i]. We first verified that the sphericity and normality assumptions of rm-ANOVA were unmet. Normality was verified with the Shapiro-Wilk normality test [29], employing the *shapiro.test* function of the *ggpubr R* package [30]. Sphericity was assessed with Mauchly's test for sphericity [31] within the *rstatix* [32] *R* package. rm-ANOVA was implemented with the *anova_test* function (*rstatix*; $p < .02$).[i] A preliminary sensitivity analysis (Table 1) executed with *G\*Power* [33] had determined that our experimental design with rm-ANOVA could identify effects of size $f = 0.218/\eta_p^2 = 0.045$ (Table 1) with a power of 90% (see Equation A in S1 Text to convert effect size of type *f* to type $\eta_p^2$). For context, we found an effect size of $f = 0.31$ (see Equations B and C in S1 Text) in our pilot study. Comparisons between both effect sizes need to be performed cautiously as the design of the rating collection has been modified between the pilot study and this report. In the contingency that at least one of the assumptions of rm-ANOVA was violated, we planned to compare two LMEs instead, one "alternative" model with defacing as a fixed effect and one "baseline" without defacing as a factor. In both models, the intercept was allowed to vary between raters (i.e., subject and rater were random effects)[j]. The models were implemented in *R* with the *lmer* function of the *lme4* package [34]. As part of regression diagnostics (e.g., non-Gaussian or heteroscedastic residuals indicate non-optimal model fit), we examined the shape of regression residuals, which are reported in the S1 Text for completeness. To test the effect of defacing, we performed a likelihood-ratio test comparing baseline and alternative LMEs. The likelihood-ratio test was implemented with the *anova* function of the *R* package *stats* [35][k]. We set the significance threshold for the likelihood ratio at $p = 0.02$. In addition, to estimate the size of the effect, we computed the noncentrality parameter $\lambda$ associated with the likelihood ratio test, which is a proxy for its power [36]. This post-hoc power analysis was then compared to the minimum power achievable from a pre-registered sensitivity analysis, which indicated that $\lambda \geq 13.017$ would allow the

**Table 1. Sensitivity analyses for rm-ANOVA and LME comparisons.** We determined that our rm-ANOVA modeling would confirm differences in manual ratings of $f = 0.218$ or larger with *G\*Power* [33]. This sensitivity corresponds to $\eta_p^2 = 0.045$ (i.e., a medium effect size) following Equation A in S1 Text. In the rm-ANOVA sensitivity analysis, we set two groups (defaced/nondefaced) and four measurements (4 raters) with a total sample size of 185 subjects from the HH site, 90% power, $\alpha = 0.02$, a nonsphericity correction of 0.34, and a correlation among repeated measures of 0.1. Note that this sensitivity analysis is conservative as we expected a much higher correlation among repeated measures, which would reduce the detectable effect size. Remaining conservative, we iteratively tried different non-sphericity correction values and kept the lowest one possible to maximize the detectable effect size. With *G\*Power* (Fig K in S1 Text), we also estimated the noncentrality parameter $\lambda$ associated with the likelihood ratio test, which is a proxy for the effect size, yielding $\lambda = 13.017$. The degrees of freedom of the likelihood ratio test correspond to the difference in parameter count between the two nested LMEs compared (see Table F in S1 Text).

| Test | Input parameters | | Output parameters | |
|---|---|---|---|---|
| rm-ANOVA | $\alpha$ err prob | 0.02 | Noncentrality parameter $\lambda$ | 13.273 |
| | Power (1 − β err prob) | 0.9 | Critical $F$ | 5.454 |
| | Total sample size | 185 | $df$ (Numerator) | 1.02 |
| | Number of groups | 2 | $df$ (Denominator) | 186.66 |
| | Number of measurements | 4 | Effect size $f$ | 0.218 |
| | Corr among rep measures | 0.1 | | |
| | Non-sphericity correction $\varepsilon$ | 0.34 | | |
| Likelihood ratio test | $\alpha$ err prob | 0.02 | Critical $\chi^2$ | 5.412 |
| | Power (1 − β err prob) | 0.9 | Noncentrality parameter $\lambda$ | 13.017 |
| | Degrees of freedom | 1 | | |

detection of differences between models at 90% power (see Table 1)$^l$. Lastly, the variance related to the intra-rater effect was estimated using the rm-ANOVA or, in the contingency case, by computing the variance of the regression coefficients linked to the random effect.

**Confirming that—on average—ratings are higher on defaced images.** We used Bland–Altman (BA) plots [37] to visualize the bias and the agreement of manual quality ratings between the nondefaced and the defaced conditions. Five BA plots were generated (Fig 2), one for each individual rater and one pooling the ratings from all raters together. We used the individual-rater BA plots to investigate whether the bias varied with respect to the quality grade attributed and how the bias changes depending on the rater.$^m$ The bias $\mathbb{E}\left[\Delta_{ndef-def}\right]$ is defined [37] as the mean of rating differences $\Delta_{ndef-def} = r^s_{ndef} - r^s_{def}$, where $r^s_{condition}$ are the ratings by one rater corresponding to the nondefaced (ndef) and defaced (def) conditions of a single subject's image $s$. Therefore, to investigate our hypothesis that humans' ratings are higher on defaced images, we confirmed that $\mathbb{E}\left[\Delta_{ndef-def}\right] < 0$ for all raters, aggregated and individually (Fig 2). Bland and Altman [37] recommend assessing the agreement between measurements by calculating the 95% limits of agreement (LoA), defined as the range in which 95% of the differences $\Delta_{ndef-def}$ are expected to fall. In practice, two measurement approaches cannot be used interchangeably if their LoA are too broad. Following [37], LoA are calculated as follows:

$$LoA = \left(\mathbb{E}\left[\Delta_{ndef-def}\right] - z_{\frac{\alpha}{2}} \cdot \sigma, \ \mathbb{E}\left[\Delta_{ndef-def}\right] + z_{\frac{\alpha}{2}} \cdot \sigma\right), \tag{1}$$

where $\sigma$ is the sample standard deviation, and $z_{\frac{\alpha}{2}}$ is the critical value—with $z_{\frac{\alpha}{2}} = 1.96$ for the 95% coverage range of the differences, and $z_{\frac{\alpha}{2}} = 2.0$ recommended as a close approximation [37]. To demonstrate that nonzero biases were not obtained by chance, we tested whether the bias was significantly negative by checking if $\Delta_{ndef-def} = 0$ was contained within the 95% confidence interval (CI) for the bias [37]. We calculated the 95% CI for the bias under the assumption of normality and reasonably large sample sizes following Bland and Altman's indications [37], with:

$$CI = \left(\mathbb{E}\left[\Delta_{ndef-def}\right] - 1.96 \cdot \frac{\sigma}{\sqrt{n}}, \ \mathbb{E}\left[\Delta_{ndef-def}\right] + 1.96 \cdot \frac{\sigma}{\sqrt{n}}\right), \tag{2}$$

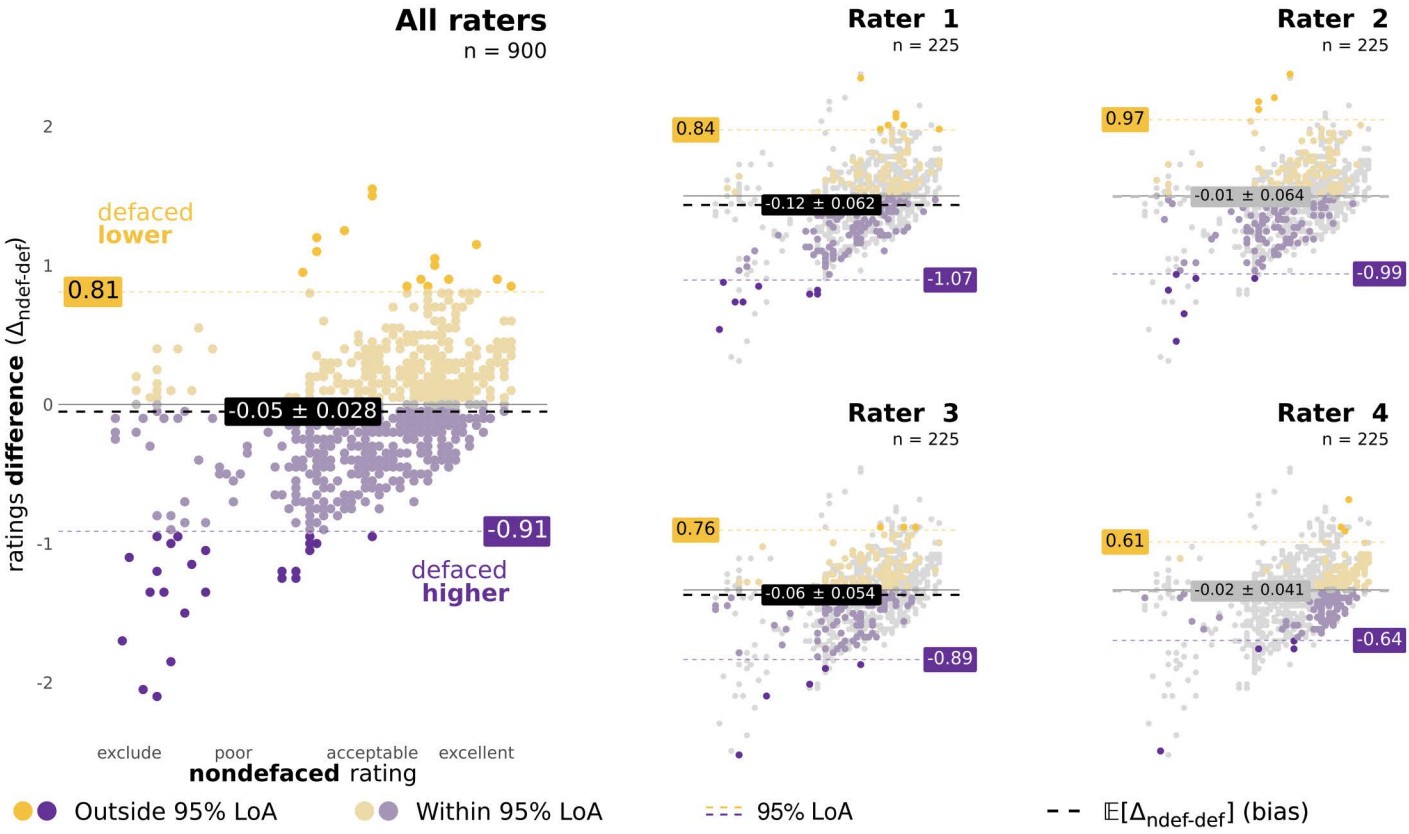

**Fig 2. Defacing biases human assessment of image quality, particularly when image quality is low.** We examined biases with an "optimized" version of the BA plot, in which the x-axis represents the rating assigned to the nondefaced version of an image. Corresponding "standard" BA plots—in which the x-axis shows the average of the two ratings [37]—are reported in Fig B in S1 Text. Rating pairs where the defaced image's quality was under-estimated with respect to the nondefaced ($\Delta_{ndef-def} > 0$) are represented in yellow. Conversely, pairs where the defaced image's quality was overesti-mated ($\Delta_{ndef-def} < 0$) are in purple. Pairs within the 95% LoA are represented with dim colors, and the LoA boundaries are indicated with dashed colored lines annotated with their value. For example, the LoA for all raters pooled together was [−0.91, 0.81] (left panel). Finally, the bias $\mathbb{E}[\Delta_{ndef-def}]$ is repre-sented by a gray or black dashed line with a label reporting value and their corresponding 95% CI interval (parametric estimation). All raters displayed 95% LoA exceeding one unit, indicating that defacing introduces large variability in human assessments. All raters had negative—albeit small—biases, indicating that they systematically rated defaced images higher. However, these biases were statistically significant only for Raters 1 and 3, as well as the four raters aggregated together—indicated by the bias label and line colored in black. Relevant statistics (bias, LoA, 95% CI) are reported in Table B in S1 Text, and the full report of statistics, including 95% CI intervals calculated both by parametric and non-parametric means are distributed within the S6 Data file. Source tabular data for the BA analysis and results in Table B in S1 Text are found within the S1 Data file.

where $n$ is the sample size. To protect from violations of the normality assumption, we also computed the CI via bootstrapping [38]. We reported both estimates and tested for the significance with the more conservative option. Additional considerations on the interpretation of BA plots, the 95% LoA, and the 95% CI of the mean bias are provided in subsection S1.1 in S1 Text.

**Determining that defacing introduces biases in *MRIQC*-generated IQMs.** Defacing impact on automatic QC was evaluated based on 58[n] out of 62 IQMs calculated by *MRIQC*. For the complete list of IQMs produced by *MRIQC* and their definitions, refer to Table 2 in [6]. A two-way repeated-measures, multivariate ANOVA (rm-MANOVA) was used to test whether defacing significantly influenced the IQMs. This test was implemented with the *multRM* function of the *MANOVA. RM* package [39] in *R*. Because many IQMs were heavily correlated (see Fig N in S1 Text), their dimensionality was reduced with principal components analysis (PCA) before applying rm-MANOVA. To ensure the defacing effects were not mitigated, PCA was applied only on the IQMs extracted from the nondefaced data, and the resulting transformation

**Table 2. The sensitivity analysis indicated that the rm-MANOVA was able to confirm differences in IQM of $f$ = 0.16 corresponding to $\eta^2$ = 0.025 (i.e., a medium effect) or greater.** We ran the sensitivity analysis with *G\*Power*, setting three groups (3 sites) and two measurements (defaced/nondefaced) with $N$ = 580 (number of T1w per subject) per condition, with 90% power, and $\alpha$ = 0.02.

| Input parameters | | Output parameters | |
|---|---|---|---|
| $\alpha$ err prob | 0.02 | Noncentrality parameter $\lambda$ | 15.5188571 |
| Power (1 − β err prob) | 0.9 | Critical F | 3.9386666 |
| Total sample size | 580 | Numerator degree of freedom | 2 |
| Number of groups | 3 | Denominator degree of freedom | 577 |
| Number of measurements | 2 | Effect size $f$ | 0.1635746 |
| | | Pillai V | 0.0260594 |

was applied to IQMs to the entire dataset. PCA was implemented with the *prcomp* function of *R*'s *stats* package, with the option *scale* = *TRUE*, meaning that the variables were standardized to have unit variance before the decomposition. The number of principal components was determined by the Kaiser criterion, which keeps components with an eigenvalue above 1.0. The rm-MANOVA was constructed with the projected IQMs as the continuous dependent variables and two categorical independent variables, one corresponding to the (nondefaced or defaced) condition of the image and the other corresponding to the scanning site. Adding the scanning site as an independent variable allowed us to control for site-effects [11,40]. We applied a significance level of $p$ < 0.02 for the rm-MANOVA and considered the $p$-values extracted under the Wald-type statistics section. A preliminary sensitivity analysis determined that our experimental design could identify, with a 90% power, effects of $f$ = 0.16/ $\eta_p^2$ = 0.025 (i.e., a medium effect) or greater with *G\*Power* (see Table 2). For context, the effect size associated with the MANOVA on the IQMs of our pilot study was $f$ = 0.16 (see Equations D and E in S1 Text for its computation). However, comparisons between both effect sizes need to be exercised with caution as the statistical designs are different. In our pilot study, we used a standard MANOVA on only five IQMs that showed the strongest bias on the BA plots. In contrast, in this pre-registration, we planned to use a repeated-measures MANOVA with all IQMs projected onto the PCA basis. In addition, the effect size reported in [5] for the *PyDeface*'s influence on IQMs ranged from $f$ = 0.045 to $f$ = 1.79, with a mean effect size across IQMs of $f$ = 0.61 (see Equation C in S1 Text for the conversion of Cohen's $d$ to Cohen's $f$). To visualize the defacing bias on the automatic quality ratings, we also generated a BA plot (as described above) for each IQM and each principal component. All BA plots are reported in Fig O through Fig R in S1 Text.

## Protocol deviations

[a] **Erroneous dataset size reported within the pre-registered plan.** The IXI dataset contains 581 images, but we erroneously stated that it had 580 T1w images in the pre-registration. This error only affected the second experiment because the reported number of participants in the HH subsample for Hypothesis 1 ($N$ = 185) was correctly reported.

[b] **Improved clarity in the description of inclusion and exclusion criteria from the IXI dataset.** Our approved Stage 1 registration left much room for precision on the inclusion and exclusion criteria. The final text clearly delineates these criteria, separated by the two hypotheses. It also resolves the problem that while we originally had proposed three hypotheses, the finally accepted Stage 1 had been reduced to the two we introduce here. Unfortunately, some text referred to hypothesis 3, which had been dismissed after peer review. This deviation also changes the location of some exclusion criteria, for example, those regarding later steps of the processing stream that are now placed at the corresponding point of the methodological description (e.g., excluding images due to insufficient quality of defacing). While the exclusion/inclusion criterion did not change, we now indicate the criterion later in the protocol.

[c] **Moved ethical statement and GDPR enforcement to more appropriate locations.** The ethical statement has been moved up to the initial paragraph of the "Data" section. The GDPR enforcement statement is now with the "Data and software availability statement".

PLOS Biology

[d] **Improved precision of the QC checkpoint after _PyDeface_.** Our approved Stage 1 should have been more specific regarding what "ineffective" defacing means. Fortunately, this exclusion criteria did not apply, and no images were excluded for this reason.

[e] **Statement of exclusions due to _MRIQC_ failures.** The approved Stage 1 plan did not explicitly anticipate the potential failure of _MRIQC_ when processing images. We report that _MRIQC_ unsuccessfully processed two subjects.

[f] Expert raters were incorporated as authors after data collection.

[g] The training session lasted 3h instead of 4h. Raters self-assessed their experience at the end of the session rather than at the beginning.

[h] The training session has been comprehensively reported, and the protocol updated correspondingly. Fig 2 in the pre-registered plan has been moved to Fig A in S1 Text.

[i] **Images rated twice for the intra-rater reliability assessment could not be fed into the planned analyses.** We had not tested at pre-registration time that neither the _rstatix_'s implementation of rm-ANOVA nor the LMEs' implementation in _lme4_ could deal with non-uniquely identified rows. As such, in the case of the images repeated twice, we excluded the ratings corresponding to the second image presentation for all raters in both conditions. We updated the preliminary sensitivity analysis for the rm-ANOVA modeling according to the reduced sample, changing the smallest detectable effect size at 90% power from $f = 0.14$ to $f = 0.218$. We have also transformed the reporting of the sensitivity analysis from the screen capture of _G*Power_ in the former Fig 3 into the new Table 1. This change had no notable consequences as we finally carried out modeling with the LMEs alternative.

[j] **Subject as a random effect in the LMEs.** Our pre-registered plan failed to identify the subject as a random effect in the LMEs, which is critical for the model to capture that the subject is a repeated measure.

[k] **Added a reference.** We added the citation to the _R_ software package [35] that was missing in the pre-registration.

[l] **Post-hoc power analysis.** Our pre-registered plan stated that "We will deem the effect irrelevant if the latter parameter is smaller than 13", which we later understood is a misinterpretation of the utility of post-hoc power analyses and discussed this in the results.

[m] **Corrected interpretation of BA plots.** Our pre-registered plan employed 95% LoA of the difference and 95% CI of the bias as interchangeable terms, which is incorrect. Our Stage 2 report was updated to employ unambiguous nomenclature for the bias $\mathbb{E}\left[\Delta_{ndef-def}\right]$, included the definition of LoA and CI as proposed initially by Bland and Altman [37,38], and provided further details regarding the interpretation of LoA and the significance of the biases.

[n] Two IQMs, Mortamet [17] "quality index 1" and the "$5^{th}$ percentile of the background intensity", were excluded because they were zero for all images. Two additional IQMs, the median and the median absolute deviation of the background intensity, were excluded as they were constantly zero except for two images. The distributions of all IQMs are plotted in the Fig L in S1 Text, with the plots corresponding to the four excluded IQMs highlighted in red.

## Methodological precisions at Stage 2 and exploratory analyses

**An alternative to the standard BA plot is presented (Hypothesis 1).** The BA plot is designed for test-retest reliability assessment, in which neither of the two measurements being presented can be assumed to be more accurate. However, in our design, ratings on the nondefaced condition can be assumed to be the reference value from which then test deviation in the defaced condition. Therefore, we modified the BA so that the _x_-axis represents nondefaced ratings (in contrast to the average between defaced and nondefaced of the "standard" plot). This modification of the BA plot displays the directionality of biases introduced by defacing more clearly. A side-by-side comparison between the standard BA plots and our "optimized" BA plot is shown in Fig B in S1 Text.

**Visually assessing homogeneity of distributions across raters (Hypothesis 1).** We visually explored the sample before carrying out the rm-ANOVA and LME tests. We evaluated the agreement across raters by generating "test–retest violin plots", where the left part of the violin shows the distribution of ratings in the nondefaced condition, and the right side shows the distribution of ratings in the defaced condition (Fig 3).

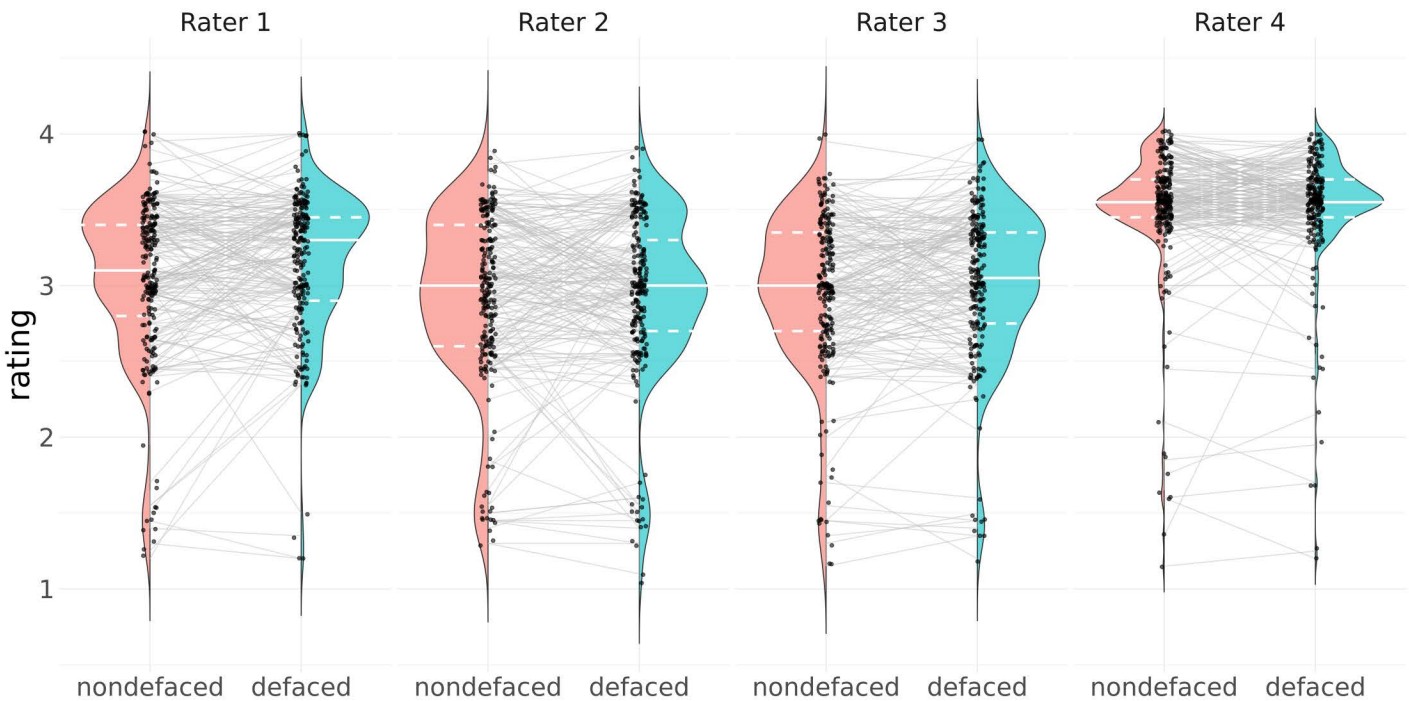

**Fig 3. Rater 4 issued visibly different ratings, generally more optimistic, than the other raters in both (defaced and nondefaced) conditions.**
The gray lines highlight the evolution of the rating between the nondefaced image and its defaced counterpart. The full white line in the violin plot represents the median of the distribution, while the dashed white lines represent the 25% and 75% quantiles. Comparing the median of the rating distribution from the nondefaced vs. defaced images, it is visible that different raters presented different bias magnitudes. Our most experienced rater (Rater 1) showed the largest bias. Rater 4's rating distribution diverged from that of the other raters, being more optimistic overall about the quality of the images. Rater 4 also displayed a lower spread in quality assessments, which translated into the narrowest 95% LoA (Fig 2). Lastly, low ratings tended to be more biased by defacing as they showed a steeper evolution line, sometimes jumping one unit or more (equivalent to switching categories in the appreciation of quality, e.g., going from "poor" to "acceptable"). Higher ratings displayed gaps within 0.5 units. BA plots support the same observation (Fig 2 and Fig B in S1 Text). Source tabular data to generate this figure are found within the S1 Data.

**Clarifications regarding the control for multiple comparisons (Hypothesis 1).** In addition to the pre-registered test, we repeated the rm-ANOVA and LMEs analyses with two subsamples. We additionally ran both models (likelihood ratio test and rm-ANOVA), including only images that were rated "poor" or "exclude" (that is, ratings below 2.45) for two reasons. First, our BA plots showed that nondefaced images that were highly rated would not increase their rating after defacing. Second, we considered that QC is more concerned with sensitivity (i.e., minimizing false negatives or, in other words, reliably detecting cases where the ratings on the defaced condition are substantially more optimistic about quality). In the second subsample, we considered only Raters 1, 2, and 3 because of the stark difference in Rater 4's distributions displayed by the test–retest violin plots (Fig 3). With these additional four tests, and considering the two tests originally planned, we corrected the $p$-values for six tests controlling for false discovery rate (FDR [41]).

**Post-hoc sensitivity analysis implementation.** Our pre-registered plan committed to estimating the effect size of the likelihood-ratio test without methodological details. The likelihood ratio test compares the ratio of likelihoods to a $\chi^2$-distribution with the degrees of freedom ($df$) equal to the difference in parameter counts between the nested alternative and baseline models [42,43]. As such, a proxy for power is given by the noncentrality parameter $\lambda$. For each LME, we obtained an unbiased estimate of the noncentrality parameter [42] using Equation F in S1 Text with $df = 1$.

**Accounting for site differences in the IQMs analysis (Hypothesis 2).** In addition to the pre-registered analysis plan, the IQMs' standardization and PCA fitting were explored site-wise instead of considering the three sites together.

We made that decision after observing that the pre-registered rm-MANOVA did not show significant IQM differences between sites (see Table 4) despite some site differences being visible in Fig M in S1 Text. Therefore, we computed the mean and standard deviation of the IQMs extracted from the nondefaced images of one site and standardized both the IQMs computed from nondefaced and defaced images from the same site by subtracting that mean and dividing by that standard deviation. Correspondingly, the following three PCA models were fit on the nondefaced sample separately by site. First, considering only IQMs computed from nondefaced images, we inspected how many components are needed in each site according to the Kaiser criterion. We chose the maximum number of components for all PCAs independently of the site. Selecting a common number of components across sites was necessary as the rm-MANOVA requires a single number of dependent variables. We then re-ran the PCA separately per site with the chosen number of components considering IQMs from both conditions. The subsequent steps of the analysis remained unchanged from the pre-registered plan.

## Results

### We confirmed defacing biased quality assessments by human raters.

A likelihood ratio test demonstrated that including defacing as a fixed effect (the alternative model) explained significantly more variance than the baseline model (see Table 3; $p_{FDR}$ = 0.0183). This result supports that defacing significantly influenced the quality assessments issued by human raters on the same images. Fig J in S1 Text shows the distribution of residuals for each fitted LME model, as well as the Q–Q plot [44] for the residuals and the subject and the rater as random effects. The residuals for both the baseline and alternative models were approximately normally distributed (see Fig J in S1 Text), with heavier left tails. The residuals distribution plot also suggested signs of heteroscedasticity in the lower ratings, although most points were still evenly distributed around zero. Table H in S1 Text reports the variance of the random effects coefficients and residuals.

We carried out the LME model comparison because the majority of the rating subgroups defined by the interaction between the defacing condition and rater identifier were non-normally distributed according to the Shapiro-Wilk normality test [29] (see Table C and Fig I in S1 Text). Nonetheless, we also carried out the planned rm-ANOVA for completeness as an exploratory analysis, and the results of the rm-ANOVA are presented in Table D in S1 Text. Even though non-normality undermines the sensitivity of rm-ANOVA, these tests also showed statistically significant biases between the defaced and nondefaced rating conditions.

Following our pre-registered plan, we also conducted a post-hoc power analysis of the alternative versus baseline LMEs comparison to estimate an effect size. We estimated the noncentrality parameter corresponding to our maximum likelihood ratio test $\lambda$ = 5.28 (see Table I in S1 Text). Therefore, $\lambda$ < 13.017 indicates that the effect size was smaller than the minimum our previous sensitivity analysis found detectable at 90% power (Table 1). Our pre-registered plan explicitly stated, "We will deem the effect irrelevant if the latter parameter [noncentrality parameter of the $\chi^2$-distribution] is smaller than 13, corresponding to the minimum power achievable from the sensitivity analysis." However, in preparing this final report, we understood these post-hoc power analyses give no basis for "deeming an effect irrelevant" (see deviation[i]), as effect size estimates can be extremely biased [45]. We report this post-hoc power analysis following the pre-registration but remark that it should be interpreted with the utmost caution.

**Table 3. The linear mixed-effect models (LME) with defacing as a fixed effect explained significantly more variance than a "baseline" counterpart without defacing.** Model comparison with a likelihood ratio test yielded a $p_{FDR}$ = 0.0183 after FDR correction. Complete reporting of the pre-registered comparison and the additional exploratory analyses are provided in Tables F and G in S1 Text.

| Sample size | $\chi^2$ | df | P(>$\chi^2$) | $p_{FDR}$ | $p_{FDR}$ < .02 |
|---|---|---|---|---|---|
| 1,480 | 6.283 | 1 | 0.0122 | 0.0183 | * |

**Table 4. Results of repeated-measures MANOVA on the projected IQMs.** We did not find an effect of the site nor an effect of defacing on the principal components extracted from the IQMs. However, applying the IQMs standardization and PCA separately per site mitigated the site effects, revealing a defacing bias. Despite being significant, the defacing bias is associated with a negligible effect size. Effect size is reported as partial $\eta^2$ ($\eta_p^2$) and was computed with the function $F\_to\_\eta^2$ from the $R$ package *effectsize* [46], which implements the formula given in Equation D in S1 Text with df = numDF and df_error = denDF. $p_{FDR}$ corresponds to $p$-values controlled for false discovery rate (FDR).

| | | Wald-type statistic (WTS) | | | | Modified ANOVA-type statistic (MATS) | | Resampling version of the tests | |
|---|---|---|---|---|---|---|---|---|---|
| | | Test statistic | df | p | $p_{FDR}$ | Test statistic | $\eta_p^2$ | $p_{WTS}$ | $p_{MATS}$ |
| Single PCA (pre-registered) | Site | 0.376 | 2 | 0.828 | 0.994 | 0.457 | $7.9 \times 10^{-4}$ | 0.829 | 0.827 |
| | Defaced | 0.003 | 1 | 0.96 | 0.96 | 0.002 | $3.5 \times 10^{-6}$ | 0.961 | 0.961 |
| Site-wise PCA (exploratory) | Site | 0.358 | 2 | 0.836 | 0.994 | 0.710 | 0.0012 | 0.833 | 0.833 |
| | Defaced | 181.182 | 1 | <.001 | <.003** | 2.726 | 0.0047 | <.001*** | <.001*** |

**Biases were larger in images rated as having lower quality in the nondefaced condition.** Our variation of the BA plot (Fig 2) and our test–retest violin plots contrasting defaced and nondefaced ratings (Fig 3) revealed that nondefaced images rated the lowest showed larger quality gaps with respect to their defaced counterpart than highly-rated nondefaced images. Low ratings presented an evolution line with a higher slope, sometimes jumping one unit (that is, a category change in the appreciation of quality, for example, from "poor" to "acceptable"), while the difference amongst the highest ratings remained within 0.5 units. Our exploratory models considering only low ratings further supported this interpretation by showing a higher effect size linked to defacing than that of the pre-registered model counterparts with all ratings (Table D in S1 Text). Although all raters displayed individual LoA larger than one unit, indicating that defacing introduced large variability, these plots also reflected that raters were not equally affected by defacing. Our most experienced rater (Rater 1) presented the highest bias $\mathbb{E}\left[\Delta_{ndef-def}\right] = -0.12$ (Table B in S1 Text), while a more junior rater (Rater 4) barely modified their quality assessment ($\mathbb{E}\left[\Delta_{ndef-def}\right] = -0.02$). The differences between nondefaced and defaced ratings by Rater 4 could only be small because most of their ratings were remarkably high as compared to the other raters (see Fig 3). Our subsample rm-ANOVA and LMEs tests without Rater 4 supported that Raters 1, 2, and 3 showed better agreement, as shown by the larger rater effect size compared to our pre-registered model counterparts with all ratings (Table D in S1 Text). $\mathbb{E}\left[\Delta_{ndef-def}\right]$ was consistently negative across all raters, implying that defaced images were, on average, scored with a higher rating. This bias was significant for Raters 1 and 3, and all raters aggregated, as shown in Fig 2 and comprehensively reported in Table B in S1 Text.

*MRIQC*-generated IQMs were mostly insensitive to defacing. Our pre-registered rm-MANOVA test (Table 4) did not reveal differences in the IQMs (S2 Data) distribution (S6 Data) before and after defacing ($F(1, 184) = 0.003$, $p = 0.96$, $\eta_p^2 = 3.5 \times 10^{-6}$) nor between sites ($F(2, 184) = 0.376$, $p = 0.828$, $\eta_p^2 = 7.9 \times 10^{-4}$). As explained in the section "Methodological precisions at Stage 2 and exploratory analyses", observing no site-effects pushed us to refine our PCA approach (S3–S5 Data). Following the Kaiser criterion, we kept 10 components for each site (eigenvalues presented in Table J in S1 Text). We also verified on a scree plot (see Fig S in S1 Text) that the components from the 11th did not explain a substantial part of the variance, confirming the choice of 10 components. The rm-MANOVA with refined PCA revealed a significant influence of defacing on the IQMs, although the effect size was very small ($F(5,14) = 8.269$, $p < 0.001$, $\eta_p^2 = 0.0047$, $f = 0.069$; see Equation A in S1 Text). In contrast, Bhalerao and colleagues [5] found bigger effect sizes for the impact of *PyDeface* on *MRIQC*-generated IQMs, with $f$ ranging from 0.045 to 1.79 and a mean $E[f] = 0.61$. These post-hoc power analyses must be interpreted cautiously for the reasons stated above regarding our corresponding analyses within the LME comparison. In a post-hoc exploration using BA plots (Fig O in S1 Text, S7 Data), we found that nearly all IQMs showcased very narrow 95% LoA and minimal biases—yet non-zero—compared to their variation ranges. The narrow LoA indicate that the variance introduced by defacing into the IQMs is relatively small. Biases were deemed significant in most

of the IQMs, which immediately follows the relatively large sample size. Only the entropy-focus criterion ("efc") emerged as an exception, showcasing a substantial bias and a broad 95% LoA, which, in addition, did not contain zero. As opposed to all other IQMs, the bias corresponding to the "efc" was also clearly noticeable in violin plots (Fig M in S1 Text).

## Discussion

Typical neuroimaging workflows involve many image-processing steps, contributing to analytical variability that cascades throughout the pipeline. Even initial processing steps considered relatively innocuous, such as reorienting the volume to align the anterior–posterior commissure line with the axial plane, may conceal relevant design decisions and inadvertently introduce variability. Because defacing has been shown to influence downstream processing [1,4,5], this registered report investigated the alterations introduced by this step in the manual and automatic QC of whole-brain T1w images. The automatic rating used image quality metrics (IQMs)—extracted with *MRIQC*—as a proxy. Our analysis for manual (human) ratings showed that a linear mixed-effects (LME) model with defacing as a fixed effect explained significantly more variability than a baseline model without defacing. However, the biases we estimated were relatively small: the largest bias $\mathbb{E}\left[\Delta_{ndef-def}\right]$ = −0.12 (Table B in S1 Text) was displayed by the most experienced rater (Rater 1). This indicates that human raters are sensitive to defacing, possibly more for more experienced raters. The first implication of this result is that QA/QC protocols establishing a visual checkpoint on the unprocessed data [23]—e.g., with *MRIQC*'s visual reports—should generate visualization and other QA/QC materials before defacing anatomical images. Defacing is generally required for compliance with applicable privacy protection regulations if data are to be processed on systems with low-security clearances or if data will be made accessible to third parties or publicly. Regardless of the motivation to deface anatomical images, our results support the recommendation of providing QA/QC materials extracted from the original (nondefaced) images alongside the defaced dataset, as long as the participant's identity cannot be reconstructed from those additional resources. This will enable data re-users to perform QA/QC in settings more consistent with those of the original authors, minimizing the risk of artifacts going unnoticed and biasing analyses.

Visual examination of the manual ratings through optimized Bland–Altman (BA) plots displayed a phenomenon we should have anticipated before pre-registration, which we acknowledge as a limitation of the present work. In agreement with our hypothesis, nondefaced images rated with the lower quality ("exclude" through "poor" and "low-acceptable" regime, from 1.0 to 2.45 on our rating scale) received substantially better ratings after defacing (see Fig 2). However, when nondefaced images were assigned high ratings (i.e., "excellent"), it was very unlikely that the defaced image received a more optimistic rating because of the scale constraint. We failed to identify this trivial limitation, and therefore, future investigations should employ datasets with balanced representations of poor-quality and excluded images. This bias is present in all datasets (as QA/QC attempts to minimize data exclusion on the grounds of insufficient quality), and it is more acute in public datasets (because data that met exclusion QC criteria are generally not shared openly as they were not considered for the analysis). For this reason, we argue that improving the automated QC of T1w images will require that all available data be shared, including samples discarded because they met the study's exclusion criteria. Doing so will enable the development of more sensitive and specific QC tools that are robust to site effects. Sharing all data is also essential because an image deemed unsuitable for one study might be valuable for another, as exclusion criteria vary based on the specific project's needs. Factors like the type of analysis or the region of interest in addressing the research question influence these QA/QC protocols [23].

While we designed the experiment with four raters, and despite the (pre-registered) efforts to calibrate the raters with a training session, one of the distributions of raters' assessments diverged from the other three raters in location and spread. Their ratings concentrated within the "acceptable-to-excellent" range. We explored all models after discarding this rater, and alongside visualization through BA plots and "test-retest violin plots", we found more reliable results (and larger effect sizes) excluding this rater. The distinctive distribution of ratings may indicate a failure in the design of the training session, leading to the rater's application of starkly different assessment criteria. Our pre-registration failed to set

corresponding QC criteria to assess the raters' performance and anticipate countermeasures. Therefore, it is critical to QA/QC protocols to assess raters' calibration before assessment and then schedule checkpoints over time to take corrective measures and identify rater disagreement early. We share all the materials in this report to allow researchers to build upon these training and calibration resources. Nonetheless, it cannot be ruled out that other circumstances determined the differences in rating distribution, such as attrition drifts (Rater 4 evaluated most of the 450 images in a single session; Fig E in S1 Text), differences in line-of-sight angle, or individual behavioral variables (e.g., sleep deprivation).

We carried out several exploratory analyses in addition to the pre-registered confirmatory tests (reported within S1 Text). For instance, we estimated, using the intra-class correlation coefficient, that our four raters presented an inter-rater reliability of 0.542 (95% CI = [0.497, 0.587]), corresponding to a moderate agreement (see subsections S2.2 and S2.3 in S1 Text). With a subsample of 40 subjects randomly selected, we ran an exploratory analysis investigating the test-retest reliability of the raters in both nondefaced and defaced conditions. This analysis revealed good intra-rater agreements (Table B in S1 Text), with varying and slight differences across raters regarding whether they became more or less critical of perceived quality over time (Figs E and F in S1 Text). Beyond identifying one rater's assessments sticking out compared to the others, our study was not designed to investigate how raters may be differently influenced by defacing (for example, in function of their experience), which remains a future line of research.

Finally, we also found that *MRIQC*'s IQMs are mostly insensitive to defacing. We tested the influence of defacing on the IQMs using rm-MANOVA after dimensionality reduction via PCA. Defacing bias became significant only when we meticulously controlled for site effects by applying standardization and PCA per site, and even then, it was associated with a minimal effect size $\eta_p^2 = 0.0047$. In a post-hoc exploration using BA plots, only the "efc" metric showed a meaningful difference between conditions out of the total 58 IQMs investigated. Although most IQMs had significant biases (Fig O in S1 Text, S7 Data) due to the large sample size, these biases were minimal, and their LoA were narrow in comparison to their corresponding variation ranges. Conversely, "efc" was the only metric displaying a broad LoA compared to its typical range (Fig O in S1 Text). In addition, except for "efc", no other IQM's LoA excluded the zero-difference line, indicating that very rarely, differences in "efc" will be small between conditions. This result seems a consequence of *MRIQC*'s definition of most metrics, such as contrast-to-noise ratio ("cnr") or signal-to-noise ratio in the gray matter ("snr_gm"), which are extracted from areas within the brain and, therefore, unaffected by defacing. The "efc" metric is an exception among IQMs, including background in their definition. For example, the "snrd_*" family of metrics is akin to their "snr_*" counterparts with the difference that the signal variability is estimated in the background [47]. While "efc" employs all the background surrounding the head and neck, "snrd_*" metrics discard background information outside of a "hat" mask that excludes the background below the nasio-occipital plane. Hence, they are also insensitive to the area altered by defacing. Our results contrast with [5], which stated that *MRIQC*-generated IQMs are significantly influenced by defacing. One explanation for the disagreement is that they evaluated different defacing methods, and *PyDeface*—our choice—yielded the smallest influence on the IQMs. Secondly, they performed univariate tests on only 10 IQMs, including "efc" and foreground-background energy ratio ("fber"), which were the IQMs varying the most with defacing when visually assessed (Fig M in S1 Text). These reasons explain why they found significant differences with a smaller sample size. Based on our findings, we recommend that *MRIQC* (and any QA/QC tool for unprocessed T1w images) include metrics sensitive to artifacts in areas typically removed or altered by defacing. Prior work by Pizarro and colleagues [48] on anatomical MRI and our later work on functional MRI [49] justified how information "outside" the brain region is relevant to assess the quality of imaging "within" the region of interest. These metrics—e.g., those proposed in [48] and derivations thereof—should be clearly annotated and excluded from analyses involving defaced images where they are not applicable. Even though *MRIQC*'s IQMs seemed insensitive to defacing, we also carried out a reliability analysis of the IQMs to determine whether defacing altered the repeatability of IQM calculation (subsection S4.4 in S1 Text), concluding that defacing did not have an appreciable impact on IQMs reliability.

This registered report lays the groundwork for a better understanding of human expert behavior in assessing anatomical T1w MRI of the human brain. It provides a tooling framework to establish a rigorous QA/QC checkpoint on the

**Table 5. Study design template.** This table summarizes the link between the hypotheses, research questions, analysis plans, sensitivity analysis, and prospective interpretation given different outcomes.

| Hypothesis | Question | Sampling plan | Analysis plan | Rationale for deciding the sensitivity of the test for confirming or disconfirming the hypothesis | Interpretation given different outcomes |
|---|---|---|---|---|---|
| Defacing influences trained raters' perception of quality | Do the quality ratings from human raters significantly vary between the defaced and the nondefaced conditions? | There is no previous analysis that can inform us on the effect size. For the rationale on how we chose the sample size, refer to the sensitivity analysis in the fifth column. | We will first verify whether the sphericity and normality assumptions of repeated-measures ANOVA (rm-ANOVA) are met. If they are, a rm-ANOVA will then be implemented in *R*. | The sensitivity analysis, reported in Table 1, indicates that at worst we will be able to confirm differences in manual ratings of $f = 0.14$ corresponding to $\eta^2 = 0.019$ (i.e., a medium effect) or greater. | $p < .02$ will indicate significance of the rm-ANOVA, thus confirming that manual quality ratings significantly vary between the defaced and nondefaced conditions. Conversely, we will interpret $p \geq .02$ as a failure to confirm our hypothesis. In any case, the post hoc power achieved and the Cohen's *f* effect size will be reported. The effect will be deemed irrelevant if the power achieved is lower than 90% or if the Cohen's *f* effect size is smaller than the minimum detectable effect size we obtained from the sensitivity analysis. |
| | | | In the contingency that at least one of the rm-ANOVA assumptions is violated, we will use linear mixed-effects models instead. To test the effect of defacing, we will perform a likelihood-ratio test comparing the models with and without adding the defaced factor as a fixed effect. | The sensitivity analysis for the likelihood ratio test is reported in Table 1. | The bias of defacing on the manual ratings will be deemed significant if the likelihood-ratio test returns $p < .02$. Conversely, we will interpret $p \geq .02$ as a failure to confirm our hypothesis. Furthermore, the effect will be deemed irrelevant if the non-centrality parameter associated with the likelihood ratio test is smaller than 13, corresponding to the minimum power achievable from the sensitivity analysis. |
| | Are ratings in the defaced condition higher than the corresponding ratings on the nondefaced condition? | | We will use Bland–Altman plots (Altman and Bland [37]) to visualize the bias and the limits of agreement of manual quality ratings between the nondefaced and the defaced condition. | | To demonstrate that the ratings of the defaced condition are higher than the corresponding ratings on the nondefaced condition, the bias should be shown to be significantly negative. A bias in the BA plot will be deemed significant if the 95% limits of agreement do not contain the zero difference. In case the 95% limits of agreement do not contain the zero difference, but the bias is positive, we will alternatively conclude that human raters perceive nondefaced images as having better quality overall. Lastly, in case the 95% limits of agreement contain the zero difference, we will conclude that we failed to verify the consistency of defacing bias on manual ratings. |
| Defacing biases automatic QA/QC of structural MRI with *MRIQC* | Do the IQMs computed by *MRIQC* significantly vary between the defaced and the nondefaced condition? | As a reference to the sensitivity analysis in the fifth column, the effect size associated with *PyDeface* influence on IQMs in (Bhalerao and colleagues [5]) ranged from $f = 0.045$ to $f = 1.79$ with a mean effect size across IQMs of $f = 0.61$. | A two-way repeated-measures MANOVA (rm-MANOVA) will be used to test whether defacing significantly influences the IQMs. However, because many IQMs are heavily correlated (see Fig N in S1 Text), we will apply PCA on the IQMs before running rm-MANOVA. | The sensitivity analysis, reported in Table 2, indicates that we will be able to confirm differences in IQM of $f = 0.16$ corresponding to $\eta^2 = 0.025$ (i.e., a medium effect) or greater. | $p < .02$ will indicate significance of the rm-MANOVA, thus confirming that the IQMs generated by *MRIQC* significantly vary between the defaced and nondefaced conditions. We will consider the *p*-values extracted under the section Wald-type statistics. Conversely, we will interpret $p \geq .02$ as a failure to confirm our hypothesis. In any case, the post hoc power achieved and the Cohen's *f* effect size will be reported. The effect will be deemed irrelevant if the power achieved is lower than 90% or if the Cohen's *f* effect size is smaller than the minimum detectable effect size we obtained from the sensitivity analysis. |

unprocessed data. Finally, this report provides the basis for standardized QA/QC protocols that are consistent across institutions, experts, and over time.

## Conclusions

The initial processing steps in the computational pipeline may have relevant downstream effects. Our report showed that defacing influences human quality assessments of T1w images, with lower-quality nondefaced images likely rated higher after defacing. Inadequate manual QC decisions deriving from the visualization of defaced images could undermine the soundness of results. *MRIQC*'s automatically extracted IQMs, on the other hand, remained largely unaffected, as the tool does not consider areas typically altered by defacing when computing the IQMs.

## Supporting information

**S1 Data. Source tabular data for the manual ratings experiment.**
(TSV)

**S2 Data. Source tabular data containing all IQMs calculated with *MRIQC*.**
(CSV)

**S3 Data. PCA projections of IQMs calculated with *MRIQC* (single standardization and PCA step).**
(CSV)

**S4 Data. PCA projections of IQMs calculated with *MRIQC* (standardization per site but a single PCA step).**
(CSV)

**S5 Data. PCA projections of IQMs calculated with *MRIQC* (standardization and PCA per site).**
(CSV)

**S6 Data. Full report of statistics, including 95% CI intervals calculated both by parametric and non-parametric means corresponding to S1 Data.**
(TSV)

**S7 Data. Full report of statistics, including 95% CI intervals calculated both by parametric and non-parametric means corresponding to S2 Data.**
(TSV)

**S8 Data. Full report of statistics, including 95% CI intervals calculated both by parametric and non-parametric means corresponding to S3 Data.**
(TSV)

**S9 Data. Full report of statistics, including 95% CI intervals calculated both by parametric and non-parametric means corresponding to S4 Data.**
(TSV)

**S10 Data. Full report of statistics, including 95% CI intervals calculated both by parametric and non-parametric means corresponding to S5 Data.**
(TSV)

**S1 Text. Supplementary Materials—Provins and colleagues (2025) "Removing facial features from structural MRI images biases visual quality assessment".** Supplementary Report containing Figs A–U, Tables A–K, and Equations A–F, completing the reporting of the main text and additional, unplanned, exploratory analyses.
(PDF)

## Acknowledgments

The Stage 1 protocol corresponding to this Registered Report was reviewed and recommended by Peer Community in Registered Reports. Table 5 reports the accepted study design template. The recommendation and review history are accessible at https://rr.peercommunityin.org/articles/rec?id=346.

## Author contributions

**Conceptualization:** Oscar Esteban.

**Data curation:** Céline Provins, Oscar Esteban.

**Formal analysis:** Céline Provins.

**Funding acquisition:** Oscar Esteban.

**Investigation:** Céline Provins, Élodie Savary, Thomas Sanchez, Emeline Mullier, Jaime Barranco, Elda Fischi-Gómez.

**Methodology:** Céline Provins, Élodie Savary, Yasser Alemán-Gómez, Jonas Richiardi, Russell A. Poldrack, Oscar Esteban.

**Project administration:** Oscar Esteban.

**Resources:** Russell A. Poldrack, Patric Hagmann, Oscar Esteban.

**Software:** Céline Provins, Élodie Savary, Oscar Esteban.

**Supervision:** Oscar Esteban.

**Visualization:** Céline Provins, Oscar Esteban.

**Writing – original draft:** Céline Provins, Oscar Esteban.

**Writing – review & editing:** Céline Provins, Élodie Savary, Thomas Sanchez, Emeline Mullier, Jaime Barranco, Elda Fischi-Gómez, Yasser Alemán-Gómez, Jonas Richiardi, Russell A. Poldrack, Patric Hagmann, Oscar Esteban.

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
