## [Editor Report · Decision Letter 0]

28 Oct 2024

Dear Dr Esteban,

Thank you for submitting your manuscript entitled "Defacing biases visual quality assessments of structural MRI" for consideration as a Stage 2 Preregistered Research Article by PLOS Biology.

Your manuscript has now been evaluated by the PLOS Biology editorial staff, as well as by an academic editor with relevant expertise, and I am writing to let you know that we would like to send your submission out for external re-review. However...

VERY IMPORTANT:

The Academic Editor has identified several issues with your submission which will need to be addressed before re-review (we understand that the issues would also have arisen at PCI RR). To this end, please login to Editorial Manager where you will find the paper in the 'Submissions Needing Revisions' folder on your homepage. Please click 'Revise Submission' from the Action Links, upload a corrected version of your manuscript, and complete all additional questions in the submission questionnaire. Here are the problems identified by the Academic Editor:

1. There are a large number of unexplained deviations between the Stage 1 and Stage 2 manuscript. The tracked-changes manuscript that you submitted makes these difficult to fully understand because large sections of text (e.g. in paragraph 1 of the introduction) are deleted and then replaced with large sections of text in which there are both similarities and differences, rather than the showing ONLY the altered text in the originally deleted sections.

Note that you are NOT free to make purely stylistic revisions between Stage 1 and 2; only changes in tense, changes to correct factual errors, and changes that reflect deviations in procedures (footnoted with explanation; see below) are permitted. Any other changes will likely need to be reverted unless they are essential for some other important reason. Those changes that are essential and substantial (e.g. in terms of changes in procedures) need to be footnoted and explained in the Stage 2 manuscript.

The AE noted that some changes are quite challenging to understand without further context, e.g. lines 101-104 in the Stage 1 manuscript describe one of the exclusion criteria, and these are altered in the corresponding section of Stage 2 without explanation. The Academic Editor thinks that you may have misunderstood the extent of leeway there is to make unnecessary changes, and that you have underestimated the degree of transparency that is required within the Stage 2 manuscript about the footnoting and explanation/justification of those changes that are necessary.

2. Please therefore replace the current version of the manuscript file with one in which all non-essential changes are reverted to the Stage 1 text. All essential changes should be listed in a separate "Protocol Deviations" section at the end of the Methods section, with superscript letters that send the reader back to specific points of deviation in the Methods - you can see an example of how this was done in a previous study here: https://journals.plos.org/plosbiology/article?id=10.1371/journal.pbio.3001566

3. Please supply a tracked-changes document that highlights additions and deletions ONLY, rather than replacing whole paragraphs with blocks of text that are very similar, which currently obscures what actual changes were made.

4. Please include the study design table (with outcomes), as this is missing.

You can see further instructions in the PCI RR guidance on Stage 2 submissions: https://rr.peercommunityin.org/help/guide_for_authors#h_97949820420921613309536944 and in the "Submitting a Stage 2 Manuscript" section of the PLOS guidelines: https://plos-marketing.s3.amazonaws.com/Marketing/Biology+Preregistered+Articles+Guidelines+for+Authors.pdf

Once your full submission is complete, your paper will undergo a series of checks in preparation for peer review. To provide the metadata for your submission, please Login to Editorial Manager (https://www.editorialmanager.com/pbiology) within four working days, i.e. by Nov 01 2024 11:59PM.

Kind regards,

Roli Roberts

Roland G Roberts PhD

Senior Editor

PLOS Biology

rroberts@plos.org

on behalf of

Christian Schnell, PhD,

Senior Editor

PLOS Biology

cschnell@plos.org

---

## [Decision Letter · Decision Letter 1]

28 Jan 2025

Dear Oscar,

Thank you for your patience while we considered your revised manuscript "Defacing biases visual quality assessments of structural MRI" for consideration as a Preregistered Research Article at PLOS Biology and apologies for the long delay in getting back to you because of delays and complications with the review process. We were only able to obtain a reviewer report from one of the previous reviewers and needed to discuss the next steps in a bit more detail. In any case, your revised study has now been evaluated by the PLOS Biology editors, the Academic Editor and one of the original reviewers.

In light of the reviews, which you will find at the end of this email, we are pleased to offer you the opportunity to address the remaining points from the reviewers in a revision that we anticipate should not take you very long. We will then assess your revised manuscript and your response to the reviewers' comments with our Academic Editor aiming to avoid further rounds of peer-review, although might need to consult with the reviewers, depending on the nature of the revisions.

We think that most of Reviewer 1's concerns can be addressed by minor clarifications and improvements to the discussion to address limitations and temper the conclusions. Please note that it is important that you do not make any unnecessary further major revisions to the existing Stage 1 content in light of these comments. Only revisions to correct factual errors and improve clarity should be made to the introduction and methods sections. The bulk of the more substantial revisions should be centered in the Discussion section.

**IMPORTANT - SUBMITTING YOUR REVISION**

*Resubmission Checklist*

*Published Peer Review*

*PLOS Data Policy*

*Blot and Gel Data Policy*

Sincerely,

Christian

Christian Schnell, PhD

Senior Editor

PLOS Biology

cschnell@plos.org

REVIEWS:

Reviewer #1 (Catherine Morgan): Defacing biases visual quality assessments of structural MRI

This work investigates if there are differences in quality ratings of T1w MRI before and after de-facing, a step commonly applied to de-identify images before wider sharing. The analysis has been performed following a PCI Registered Report.

The work seems to have been carried out in accordance with the pre-registered methodology and any deviations clearly documented and results mainly well presented.

I think that the manuscript should be published but suggest a few additions and changes that I think might improve the readability and help readers understand the findings more generally.

My take home from the findings is that there are no differences in automated QC measures (using MRIQC) between de-faced and non-defaced images. Except for (@L535 ) entropy-focus criterion. I think to help readers understand this finding, reiterate the difference in efc is only 1 IQM out of <X total no.> of QC measures derived and tested. And provide more on what some of the other IQMs are that are not different between the defaced and non-defaced conditions - i.e. I would imagine snr_gm and cnr are likely to be more important QC measures to most readers - so some context here would help.

And if essentially there is no difference in automated QC measures by defacing, I don't agree with the recommendation in the Discussion starting on L481 "Since defacing…" which says it's important to report results from the non-defaced data? - but why, when it's essentially no different to the de-faced data? If this specifically relates to the manual rating please see comment on this below.

In the concluding sentences it's stated that you've shown how defacing can "degrade data reusability" but I'm not sure you have? And again I don't think the conclusion that "We argue that sharing QA/QC materials from nondefaced images can help mitigate these issues while ensuring participant privacy" is the correct conclusion based on the results which show that for automated QC defacing doesn't impact results.

For the manual ratings - my understanding is that the raters saw the whole image, so it's immediately obvious if the image is a defaced one or not? Wouldn't it be the case then that the raters, end up focusing on IQ issues related to those around the face - e.g. ghosting from the eyes - but really what's important is the IQ of the brain? Making the assumption that most research will be using the brain data. And then probably skull stripping in their analysis.

Then maybe focusing on the IQ of the brain becomes overlooked and raters, consciously or not, focus on the face area? What I'm wondering is that maybe non-defaced images are simply marked to have worse quality due to eye motion, but that's not really a good measure of overall IQ for researchers who want to study the brain. This would be easy to check - if the raters mainly specified "eye spillover…" in the rating widget when the de-faced images were scored better? I think this should be reported and discussed. And if this is true, I think findings about differences in manual QC should be down-played a little.

Minor points

"QA/QC" used throughout - then sometimes QA and sometimes QC on its own. Are they intended to be used interchangeably or as separate things - if interchangeably only one term is needed, if separate, explain.

Assuming "eye-spillover" is ghosting from eye motion?

I don't follow this statement - and it makes it sound like the main result has been caused by poor study design? I think it needs a better explanation

L331 "Our variation of the BA plot revealed that images rated highly in the nondefaced condition were, by design, very unlikely to receive a higher rating in the defaced condition, which we did not anticipate at pre-registration time."

Explain how your "Background noise" image in Figure 1 has been made (appears to be by thresholding but should be included in the caption)

Include a comment at the start of the data section L92 that the lettered footnotes are related to protocol deviations which can be found at the end.

The bias - defined L421 - should be defined on first use not after, i.e. before L415 - and a quick note on how it's computed (assuming it is the average of non-deface- defaced score for all images that rater scored?)

Figure 2 - "Defaced Underestimated" label would be clearer as "Defaced rated worse" (or "lower"), and similarity - "Defaced overestimated" might be clearer as "Defaced rated better" (or "higher").

Abstract - L17 - "requirement" should be "step" since requirement might be different for dependent on the study.

L39 - what type of automated analysis method fail? and what does "failed in execution" mean? Grammatical error in there too.

L41 - outcomes of interest such as? GM volume? Give example

L57 - replace "is" with "may be"

L93 "confirmatory" here maybe lessens the importance of the work carried out?

L108 PyDeface because it <currently> presents the highest success rate…

L111 the sentence starting "Thus…" could be reworded to avoid double negatives and be easier to understand. Subsequently the sentence on L113 starting "Under our hypotheses … could be re-worded as I don't quite follow what is trying to be said

IQM seems to be first defined in the conclusion

---

## [Editor Report · Decision Letter 2]

27 Mar 2025

Dear Oscar,

Thank you for your patience while we considered your revised manuscript "Defacing biases visual quality assessments of structural MRI" for publication as a Preregistered Research Article at PLOS Biology. This revised version of your manuscript has been evaluated by the PLOS Biology editors and the Academic Editor.

Based on our Academic Editor's assessment of your revision, we are likely to accept this manuscript for publication, provided you satisfactorily address the following data and other policy-related requests:

* We would like to suggest a different title to improve its accessibility for our broad audience: ""Removing facial features from structural MRI images biases visual quality assessment"

* Please include the approval/license number of the ethical approval for experiments (if required) and include information in the Methods section whether the study has been conducted according to the principles expressed in the Declaration of Helsinki. Please also specify whether the participants provided written or oral consent.

* DATA POLICY:

Regardless of the method selected, please ensure that you provide the individual numerical values that underlie the summary data displayed in the following figure panels as they are essential for readers to assess your analysis and to reproduce it: 3, S3, S5, S6, S12, S13 and S20.

* CODE POLICY

* Please move all references from the supplementary information to the main reference list.

* Please include a link to the OSF repository where the Stage 1 IPA manuscript can be found.

* Please mention somewhere in the manuscript (e.g. Acknowledgements) that the Stage 1 submission was reviewed and recommended by Peer Community in Registered Reports, with a link to the public recommendation and review history: https://rr.peercommunityin.org/articles/rec?id=346

We expect to receive your revised manuscript within two weeks.

*Published Peer Review History*

*Press*

Sincerely,

Christian

Christian Schnell, PhD

Senior Editor

cschnell@plos.org

PLOS Biology

---

## [Editor Report · Decision Letter 3]

2 Apr 2025

Dear Oscar,

Thank you for the submission of your revised Preregistered Research Article "Removing facial features from structural MRI images biases visual quality assessment" for publication in PLOS Biology. On behalf of my colleagues and the Academic Editor, Christopher Chambers, I am pleased to say that we can in principle accept your manuscript for publication, provided you address any remaining formatting and reporting issues. These will be detailed in an email you should receive within 2-3 business days from our colleagues in the journal operations team; no action is required from you until then. Please note that we will not be able to formally accept your manuscript and schedule it for publication until you have completed any requested changes.

PRESS

Sincerely, 

Christian

Christian Schnell, PhD

Senior Editor

PLOS Biology

cschnell@plos.org